# Learn from Your Mistakes: Tree-like Self-Play for Secure Code LLMs

**Wenqi Chen** [* 1] **Ziyan Zhang** [* 1] **Bin Wang** [2 †] **Lin Liu** [3 †] **Hengheng Zhang** [4] **Zhengsu Chen** [5]

## Abstract

While Large Language Models (LLMs) excel in code generation, they remain prone to replicating subtle yet critical vulnerabilities endemic to their training data. Current alignment techniques, such as Supervised Fine-Tuning (SFT) and Reinforcement Learning (RL), typically apply coarse-grained optimization at the sequence level. This approach often fails to address the localized nature of security flaws, where a single incorrect token choice can compromise an entire program. To bridge this gap, we introduce **T**ree-like **S**elf-**P**lay (**TSP**), a framework that reframes secure code generation as a fine-grained sequential decision process. Unlike standard methods that blindly maximize likelihood, TSP constructs a decision tree where the model explores branching trajectories—generating both secure "golden paths" and vulnerable variants. By treating code generation as a self-play game, the model learns to strictly discriminate against its own localized errors. This provides a dense, on-policy learning signal that forces self-correction precisely at the critical decision nodes where vulnerabilities typically emerge. Our experiments demonstrate that TSP fundamentally enhances model reliability. In Python security benchmarks, TSP boosts CodeLlama-7B's pass rate (SPR@1) to 75.8%, significantly outperforming SFT (57.0%) and unstructured self-play baselines. Crucially, TSP induces robust out-of-distribution generalization: the model not only reduces vulnerabilities in unseen categories (CWEs) by 24.5% but also successfully transfers security principles learned from C/C++ to diverse languages, includ-

ing Python, Go, and JavaScript. This suggests that TSP does not merely memorize patches, but internalizes abstract, language-agnostic security logic. Our code and data can be found at `https://github.com/Easonnoway/TSP`.

## 1. Introduction

The integration of LLMs into software engineering has revolutionized the development lifecycle (Li et al., 2022; Ouyang et al., 2022). However, this reliance on automated generation introduces critical security risks (Negri-Ribalta et al., 2024; Jamdade & Liu, 2024; Perry et al., 2023; Wang et al., 2025b). Since LLMs are trained on vast open-source corpora containing latent vulnerabilities, they inadvertently propagate insecure patterns—ranging from SQL injections to deprecated APIs. Consequently, enhancing the security awareness of these models remains a paramount challenge.

Current mitigation strategies primarily adapt general alignment paradigms but face significant limitations in the code domain. SFT (Ouyang et al., 2022; Chung et al., 2024) optimizes entire sequences, diluting the learning signal across the code block and failing to pinpoint specific erroneous tokens (Li et al., 2022; Luo et al., 2024; Wei et al., 2024). While RL approaches offer dynamic guidance (Le et al., 2022; Gehring et al., 2024; Dou et al., 2024), they typically rely on program-level feedback, which is too coarse to correct localized decision errors. Furthermore, robust security alignment is hindered by three persistent obstacles: (1) the sample inefficiency of traditional methods (Ji et al., 2024); (2) the scarcity of high-quality, vulnerability-specific datasets (Fan et al., 2020; Croft et al., 2023; He et al., 2024; Lian et al., 2025); and (3) the lack of *on-policy* negative examples, which limits the model's ability to distinguish between plausible but insecure alternatives and secure solutions (Tang et al., 2024; Tajwar et al., 2024; He & Vechev, 2023; Hajipour et al., 2024; Zhang et al., 2024).

To transcend these limitations, we posit that mastering secure coding requires identifying and correcting errors at their specific locus of origin. We introduce **T**ree-like **S**elf-**P**lay (**TSP**), a novel training framework that frames code generation as a tree traversal. We identify "CWE risk nodes"—critical forks where insecure choices lead to

---

[*]Equal contribution [†]Corresponding authors. [1]University of Electronic Science and Technology of China, Chengdu, China [2]Peking University, Beijing, China [3]University of Science and Technology of China, Hefei, China [4]Hefei University of Technology, Hefei, China [5]Beihang University, Beijing, China. Correspondence to: Bin Wang <thebinking66@stu.pku.edu.cn>, Lin Liu <ll0825@mail.ustc.edu.cn>.

*Proceedings of the $43^{rd}$ International Conference on Machine Learning*, Seoul, South Korea. PMLR 306, 2026. Copyright 2026 by the author(s).

vulnerabilities. At these nodes, TSP institutes an adversarial self-play mechanism (Figure 1). The model utilizes its own exploratory, insecure branches as an "opponent," learning to distinguish the secure "golden path" from these self-generated flaws.

TSP offers distinct advantages over existing paradigms:

1. **Data Efficiency via Self-Play:** By autonomously generating on-policy positive and negative examples, TSP eliminates the reliance on expensive human annotations and sparse vulnerability datasets.

2. **Granular, Hierarchical Feedback:** The tree representation mirrors the nested structure of real code, enabling the model to learn from both coarse- and fine-grained signals at each abstraction level (Qiu et al., 2024; Hou et al., 2025).

3. **Performance and Generalization:** Extensive experiments demonstrate that TSP can boost security pass rate of model from 57.0% to 75.8%. Crucially, the model exhibits strong out-of-distribution generalization, effectively reducing vulnerabilities in unseen CWE categories by 32% and transferring security principles from C/C++ to diverse languages.

## 2. Methodology

### 2.1. Problem Setting: Secure Code Generation

We formulate secure code generation as a conditional language modeling task. Given a prompt $\boldsymbol{x}$, the model $\pi_{\boldsymbol{\theta}}$ generates a code sequence $\boldsymbol{y} = (y_1, y_2, \ldots, y_T)$ by computing the product of conditional probabilities:

$$p_{\boldsymbol{\theta}}(\boldsymbol{y}|\boldsymbol{x}) = \prod_{t=1}^{T} p_{\boldsymbol{\theta}}(y_t|\boldsymbol{x}, y_{<t}) \tag{1}$$

where $y_{<t}$ denotes the prefix tokens. This process can be viewed as traversing a generation tree, where each $y_t$ represents a branching decision.

Standard SFT optimizes the log-likelihood of a secure "golden" dataset $\mathcal{D}$:

$$\mathcal{L}_{\text{SFT}}(\boldsymbol{\theta}) = \mathbb{E}_{(\boldsymbol{x},\boldsymbol{y})\sim\mathcal{D}}\left[\log p_{\boldsymbol{\theta}}(\boldsymbol{y}|\boldsymbol{x})\right] \tag{2}$$

While effective for syntax, SFT lacks granularity for security. It reinforces the entire sequence $\boldsymbol{y}$ uniformly, failing to isolate specific, secure-critical tokens (e.g., input sanitization) from the rest of the functional code.

RL typically aligns the model by optimizing an expected reward $R(\boldsymbol{y})$ with a KL-divergence constraint to maintain coherence with a reference policy $\pi_{\text{ref}}$:

$$J(\boldsymbol{\theta}) = \mathbb{E}_{\boldsymbol{y}\sim\pi_{\boldsymbol{\theta}}}[R(\boldsymbol{y})] - \beta D_{KL}\left(\pi_{\boldsymbol{\theta}}||\pi_{\text{ref}}\right) \tag{3}$$

However, standard RL suffers from the *credit assignment problem*. Since rewards are sparse and computed only upon program completion, the feedback fails to pinpoint the precise locus of a vulnerability. For instance, if a model chooses the unsafe `strcpy` over `strncpy`, a low sequence-level reward $R(\boldsymbol{y})$ does not explicitly inform the model which token caused the vulnerability.

To address this, our **TSP** framework shifts the optimization focus from global sequence rewards to critical *decision nodes* within the generation tree, providing dense, token-level feedback where vulnerabilities originate.

### 2.2. Modeling Vulnerabilities as Divergences at Risk Nodes

Following the problem setting, the generation of a code snippet $\boldsymbol{y}$ from a prompt $\boldsymbol{x}$ can be visualized as a path through a generation tree, $\mathcal{T}(\boldsymbol{x})$. The root of the tree is the initial prompt, and each subsequent node $v$ corresponds to a unique prefix of the code, denoted as $\boldsymbol{y}_{<t_v}$, representing the token sequence generated up to step $t_v$. A complete program corresponds to a full path from the root to a leaf node. In this context, a security vulnerability can be pinpointed to a specific decision point. We term these critical junctures *CWE Risk Nodes*. Consider the task of copying a string in C and an unsafe model, as shown in Table 1.

While TSP applies gradient updates at the token level (optimizing the logits of a specific generation step), the identification and contextualization of a CWE Risk Node are inherently semantic. Real-world vulnerabilities rarely manifest as isolated token errors; they are often the culmination of complex data-flow or control-flow logic. To capture this complexity, TSP leverages the advanced semantic reasoning capabilities of large language models during the automated annotation pipeline (as detailed in Section 3.2). Rather than relying on superficial token matching or rigid heuristics, the annotator LLM analyzes the entire function's context—evaluating control structures, variable scoping, and specific CWE definitions—to isolate the precise root cause within multi-line program logic.

Embodying the wisdom of the adage, *"a fall into a pit, a gain in your wit"*, TSP leverages this insight by concentrating its contrastive learning objective exclusively on these identified CWE risk nodes. This process forces the model to distinguish the secure 'golden path' from locally divergent, insecure code generations, thus turning potential failures into learned wisdom.

### 2.3. The TSP Optimization Framework

The core of TSP is a self-play game on the generation tree. It involves two players derived from the same LLM: an *opponent player*, $p_{\boldsymbol{\theta}_t}$, from iteration $t$, and a *main player*,

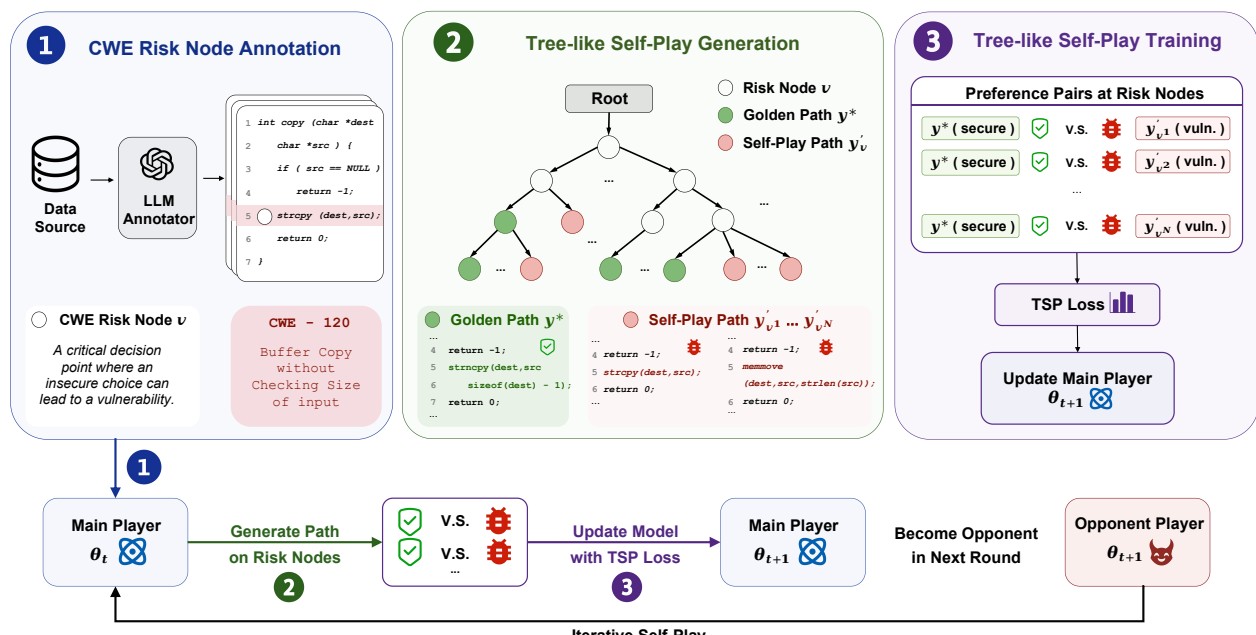

*Figure 1.* Overview of the Tree-like Self-Play framework. ***Step 1***: CWE Risk Node Annotation. An LLM annotator identifies critical risk nodes where vulnerabilities originate. ***Step 2***: Tree-like Self-Play Generation. The model generates insecure self-play paths alongside the secure golden path at these nodes. ***Step 3***: Tree-like Self-Play Training. The main player optimizes preference pairs via TSP loss, then acts as the opponent for the next iteration.

*Table 1.* Definitions of Paths and Nodes in CWE Context

| Concept | Description |
|---|---|
| *Golden Path* $\boldsymbol{y}^*$ | A secure code path where the model generates tokens for the safe function `strncpy(dest, src, sizeof(dest)-1);` given prompt $\boldsymbol{x}^*$. |
| *CWE Risk Node* $v$ | A critical juncture corresponding to the prefix immediately before the function name selection. The token at step $t_v$ is decisive for preventing CWE-120 (*Buffer Copy without Checking Size of Input*). |
| *Self-Play Path* $\boldsymbol{y}'_v$ | An alternative path generated by the unsafe model. At node $v$, the model outputs the vulnerable function `strcpy(dest, src);`, leading to a security flaw. |

$p_{\boldsymbol{\theta}}$, which is being optimized. The model learns from data generated by its past self, rather than a competitive zero-sum game.

For each ground-truth sample $(\boldsymbol{x}, \boldsymbol{y})$, we identify the set of pre-annotated CWE Risk Nodes $\mathcal{V}_{\text{risk}}(\boldsymbol{y})$ on its golden path. For each risk node $v \in \mathcal{V}_{\text{risk}}(\boldsymbol{y})$, corresponding to prefix $\boldsymbol{y}_{<t_v}$, we use the opponent player $p_{\boldsymbol{\theta}_t}$ to generate the sequence $\boldsymbol{y}'_v$. This sequence is identical to $\boldsymbol{y}$ up to the prefix $\boldsymbol{y}_{<t_v}$ but diverges afterward.

The objective is to train the main player $p_{\boldsymbol{\theta}}$ to assign a higher score to the golden path $\boldsymbol{y}$ than to any self-play path $\boldsymbol{y}'_v$. We use a convex, monotonically decreasing loss function $\ell(z) := \log(1 + \exp(-z))$ to prevent the excessive growth in the absolute value of scoring function $f(\boldsymbol{x}, \boldsymbol{y})$. The overall objective is:

$$\boldsymbol{\theta}_{t+1} = \arg\min_{\boldsymbol{\theta} \in \boldsymbol{\Theta}} \mathcal{L}_{\text{TSP}}(\boldsymbol{\theta}, \boldsymbol{\theta}_t) \tag{4}$$

The overall objective function $L_{\text{method}}$ is formulated as the expectation of a sample-wise loss function over the training data distribution $\mathcal{D}$. Specifically, for a given sample $(\boldsymbol{x}, \boldsymbol{y}) \sim \mathcal{D}$, the loss is calculated hierarchically by averaging the individual losses across all its corresponding risk nodes $v \in \mathcal{V}_{\text{risk}}(\boldsymbol{y})$. The complete objective is defined as:

$$\mathcal{L}_{TSP} = \mathbb{E}_{(\boldsymbol{x}, \boldsymbol{y}) \sim \mathcal{D}} \left[ \frac{1}{|\mathcal{V}_{\text{risk}}(\boldsymbol{y})|} \sum_{v \in \mathcal{V}_{\text{risk}}(\boldsymbol{y})} \mathcal{L}_v(\boldsymbol{x}, \boldsymbol{y}; \boldsymbol{\theta}_t) \right] \tag{5}$$

where $\mathcal{L}_v(\boldsymbol{x}, \boldsymbol{y}; \boldsymbol{\theta}_t)$ represents the loss associated with a single risk node $v$, and $|\mathcal{V}_{\text{risk}}(\boldsymbol{y})|$ is the total number of risk nodes for the sample label $\boldsymbol{y}$. This formulation ensures that our optimization process accounts for the multi-faceted risk

**Algorithm 1** Tree-like Self-Play

---

**Input:** SFT Dataset with annotated risk nodes $\{(\boldsymbol{x}_i, \boldsymbol{y}_i, \mathcal{V}_{\text{risk}}(\boldsymbol{y}_i))\}_{i=1}^N$, initial LLM $p_{\boldsymbol{\theta}_0}$, iterations $T$.

**for** $t = 1, \ldots, T$ **do**
  Initialize an empty set of comparison pairs $\mathcal{P}_t = \emptyset$.
  **for** $i = 1, \ldots, N$ **do**
    Let the ground-truth response be $\boldsymbol{y}_i$.
    **for** each *CWE Risk Node* $v \in \mathcal{V}_{\text{risk}}(\boldsymbol{y}_i)$ **do**
      Let $k_v$ be the token index of node $v$ in $\boldsymbol{y}_i$.
      Generate $\boldsymbol{y}'_{i,v} \sim p_{\boldsymbol{\theta}_{t-1}}(\cdot|\boldsymbol{x}_i, \boldsymbol{y}_{i,<k_v})$.
      Add the self-play pair $(\boldsymbol{y}_i, \boldsymbol{y}'_{i,v})$ to $\mathcal{P}_t$.
    **end for**
  **end for**
  Update parameters $\boldsymbol{\theta}_t$ via Eq. (8) over $\mathcal{P}_t$.
**end for**
**Output:** Optimized parameters $\boldsymbol{\theta}_T$.

---

structure inherent in the data. where the loss for a single risk node $v$ is:

$$\mathcal{L}_v(\boldsymbol{x}, \boldsymbol{y}; \boldsymbol{\theta}_t) = \mathbb{E}_{\boldsymbol{y}'_v \sim p_{\boldsymbol{\theta}_t}(\cdot|\boldsymbol{x}, \boldsymbol{y}_{<t_v})} \left[ \ell\big( f(\boldsymbol{x}, \boldsymbol{y}) - f(\boldsymbol{x}, \boldsymbol{y}'_v) \big) \right] \tag{6}$$

In practice, for computational efficiency, the expectation $\mathbb{E}$ in Eq. (6) is approximated via a single Monte Carlo sample, as detailed in Algorithm 1. Following DPO (Rafailov et al., 2023), we define the scoring function $f$ as the scaled log-likelihood ratio:

$$f(\boldsymbol{x}, \boldsymbol{y}) = \lambda \log \frac{p_{\boldsymbol{\theta}}(\boldsymbol{y}|\boldsymbol{x})}{p_{\boldsymbol{\theta}_t}(\boldsymbol{y}|\boldsymbol{x})} \tag{7}$$

where $\lambda$ is a scaling factor that controls the strength of the preference update, its value is determined empirically. Intuitively, this scoring function measures how much the main player's policy has improved relative to the opponent's fixed policy. A positive score indicates the main player is more likely to generate the sequence than the opponent. This formulation elegantly bridges preference learning with the generative task of updating the LLM.

### 2.4. The Iterative Update Process

The training of TSP proceeds through a series of self-play iterations. The iterative cycle consists of three key steps:

1. **Generation**: The fixed opponent player $p_{\boldsymbol{\theta}_t}$ generates self-play sequences as on-policy negative data at each risk node.

2. **Learning**: The main player $p_{\boldsymbol{\theta}}$ is trained using the collected preference pairs (golden path vs. self-play path) to minimize the TSP loss.

3. **Update**: Once training for the round is complete, the main player's parameters are used to update the opponent for the next iteration: $\boldsymbol{\theta}_t \leftarrow \boldsymbol{\theta}_{t+1}$.

The process to train the main player $\boldsymbol{\theta}_{t+1}$ is to optimize following loss function over a batch of $N$ samples :

$$\underset{\boldsymbol{\theta} \in \Theta}{\arg\min} \, \mathbb{E}_{\substack{\boldsymbol{y}'_v \sim p_{\boldsymbol{\theta}_t}(\cdot|\boldsymbol{x}, \boldsymbol{y}_{<t_v}) \\ (\boldsymbol{x},\boldsymbol{y}) \sim \mathcal{D} \\ v \sim \mathcal{V}_{\text{risk}}(\boldsymbol{y})}} \left[ \ell\Bigg( \lambda \log \frac{p_{\boldsymbol{\theta}}(\boldsymbol{y}_i|\boldsymbol{x}_i)}{p_{\boldsymbol{\theta}_t}(\boldsymbol{y}_i|\boldsymbol{x}_i)} \right.$$
$$\left. - \lambda \log \frac{p_{\boldsymbol{\theta}}(\boldsymbol{y}'_{i,v}|\boldsymbol{x}_i)}{p_{\boldsymbol{\theta}_t}(\boldsymbol{y}'_{i,v}|\boldsymbol{x}_i)} \Bigg) \right] \tag{8}$$

where $\boldsymbol{y}'_{i,v}$ is the self-play code generation for sample $i$ generated at risk node $v$. The full iterative process allows the model to progressively improve by learning to correct the more subtle errors its previous self was still making.

As mentioned in SPIN (Chen et al., 2024), the full iterative process can be summarized as:

$$\cdots \rightarrow \underbrace{p_{\boldsymbol{\theta}_t}(\cdot|\mathbf{x})}_{\substack{\text{Fixed Opponent} \\ \text{Generates } \{\mathbf{y}'_v\}}} \rightarrow \underbrace{\boldsymbol{\theta}_{t+1} = \underset{\boldsymbol{\theta}}{\arg\min} \, L_{\text{TSP}}(\boldsymbol{\theta}, \boldsymbol{\theta}_t)}_{\substack{\text{Main Player Training} \\ \text{via Eq. (8)}}}$$
$$\rightarrow \underbrace{p_{\boldsymbol{\theta}_{t+1}}(\cdot|\mathbf{x})}_{\substack{\text{New Opponent} \\ \text{for next iteration}}} \rightarrow \cdots$$

### 2.5. Analysis of Node-based Optimization and Convergence

The optimization dynamic of TSP is driven by the gradients derived from the loss function. For a single data point $(\boldsymbol{x}, \boldsymbol{y})$ and $\boldsymbol{y}'_v$, the gradient of the inner loss term is:

$$\nabla_{\boldsymbol{\theta}} \big( f(\boldsymbol{x}, \boldsymbol{y}) - f(\boldsymbol{x}, \boldsymbol{y}'_v) \big) = \lambda \Big( \nabla_{\boldsymbol{\theta}} \log p_{\boldsymbol{\theta}}(\boldsymbol{y}|\boldsymbol{x})$$
$$- \nabla_{\boldsymbol{\theta}} \log p_{\boldsymbol{\theta}}(\boldsymbol{y}'_v|\boldsymbol{x}) \Big) \tag{9}$$

In the context of policy gradient methods, the total gradient of the loss function $L_{\text{TSP}}$ is formulated as an expectation over all samples and their corresponding risk nodes. The overall gradient is given by:

$$\nabla_{\boldsymbol{\theta}} L_{TSP} = \mathbb{E} \left[ \frac{\lambda}{|\mathcal{V}_{\text{risk}}(\boldsymbol{y})|} \sum_{v \in \mathcal{V}_{\text{risk}}(\boldsymbol{y})} \ell'(\cdot) \cdot \boldsymbol{g}_v(\boldsymbol{\theta}) \right] \tag{10}$$

Here, the term $\boldsymbol{g}_v(\boldsymbol{\theta})$ represents the standard score function gradient at a specific risk node $v$:

$$g_v(\boldsymbol{\theta}) \triangleq \nabla_{\boldsymbol{\theta}} \log p_{\boldsymbol{\theta}}(\boldsymbol{y}|\boldsymbol{x}) - \nabla_{\boldsymbol{\theta}} \log p_{\boldsymbol{\theta}}(\boldsymbol{y}'_v|\boldsymbol{x}) \qquad (11)$$

**Convergence Properties:** The structure of this gradient provides a more stable and effective learning signal.

1. **Reduced Gradient Variance:** The set of self-play paths $\{\boldsymbol{y}'_v\}$ are structurally related to the positive sample $\boldsymbol{y}$, as they share long common prefixes. Averaging the gradients over these high-signal, closely-related pairs provides a more stable estimate of the true gradient direction compared to using a single, potentially noisy program-level reward.

2. **Targeted and Efficient Updates:** This is the principal advantage of TSP as the gradient is computed *only* from comparisons at critical risk nodes. This focuses the entirety of the optimization pressure on fixing potential security flaws, rather than diluting the learning signal across hundreds of syntactically correct but security-irrelevant tokens.

This node-wise supervision signal guides the optimization towards a more robust convergence where the model is not only globally correct but also locally secure at each critical generation step.

## 3. Dataset Construction

The efficacy of the proposed TSP framework hinges on the availability of granular security insights, surpassing the limitations of traditional binary labels of "secure" or "vulnerable." To drive the model's self-correction mechanism, it is imperative to pinpoint CWE Risk Nodes—critical decision points within secure code where vulnerabilities are conceptually liable to emerge. To this end, we constructed a high-quality, customized research dataset through the systematic annotation and rigorous validation of a large-scale real-world corpus.

### 3.1. Data Source and Filtering

We ground our dataset construction in DiverseVul, a comprehensive C/C++ vulnerability database chosen for its exceptional diversity. DiverseVul aggregates 18,945 vulnerable and 330,492 non-vulnerable functions across 797 distinct projects, spanning 150 CWEs categories. Derived from vulnerability-fixing commits, it authentically captures real-world coding paradigms and security flaws.

From this extensive pool, we utilized the patched, secure versions of the code as the ground truth baseline. We filtered the data to extract a representative corpus of 1,353 secure function samples, each associated with explicit CWE classification information. This curated subset provides a manageable yet statistically significant foundation for fine-grained risk node annotation.

### 3.2. Automated Annotation of CWE Risk Nodes via LLMs

The core of our data preparation is the automated identification of CWE Risk Nodes. We define a Risk Node as a specific code location that represents a potential vulnerability trigger, even within syntactically correct secure code. For example, in a function using `strncpy` to prevent buffer overflows, the invocation itself is a Risk Node for CWE-121, as a less rigorous implementation might default to the unsafe `strcpy`.

To systematically extract these nodes, we implemented an automated pipeline driven by LLMs utilizing a structured prompt design. This design enforces two key constraints: strict formatting to ensure machine-readable output for downstream training, and a criticality principle that compels the model to isolate the specific line representing the root cause or direct entry point within multi-line vulnerabilities, thereby eliminating redundancy. Full details on the prompt templates and guidelines are provided in Appendix C.

### 3.3. Annotation Quality Validation

Given that the reliability of TSP depends on accurate risk identification, we instituted a multi-stage validation protocol to ensure data quality.

First, we constructed a golden-standard validation set using a human cross-validation methodology. We randomly sampled 15% of the dataset for manual annotation by two independent security experts following the same guidelines as the LLM. The inter-rater reliability between experts yielded a Cohen's Kappa ($\kappa$) of 0.89, confirming that our definition of Risk Nodes is unambiguous and reproducible.

Second, we compared the LLM-generated annotations against this human-verified golden standard (with disagreements adjudicated by a senior expert). The automated pipeline achieved a $\kappa$ coefficient of 0.86 relative to human consensus. This high alignment demonstrates that our automated pipeline meets the rigorous standards required for research-grade security datasets.

## 4. Experiments

To rigorously evaluate the effectiveness of our proposed TSP approach, we conducted a series of experiments designed to investigate three key research questions (RQs):

- **RQ1:** Does our TSP method significantly improve the security of code generated by LLMs across different

programming languages compared to baseline?

- **RQ2:** How well do the security enhancements from TSP generalize across programming languages?

- **RQ3:** How well do the security enhancements from TSP generalize to unseen CWEs?

### 4.1. Experimental Setup

**Base Models and Datasets**  Our experimental framework is built upon three open-source LLMs: CodeLlama-7B, Qwen2.5-Coder-7B, and Qwen2.5-Coder-3B. To ensure a comprehensive evaluation of security hardening, our methodology utilizes several specialized datasets tailored to specific tasks. For fine-tuning and evaluation in Python, we employ the training set from the original SafeCoder model and perform the final assessment on the SecurityEval benchmark, which consists of 121 security-centric programming prompts. For C/C++ experiments, the extensive DiverseVul dataset serves as a unified source for both training and testing samples. Specifically, to evaluate CWE generalization (RQ3), the training partition is curated to cover 110 distinct CWE types, while the corresponding test set contains 150 samples representing 40 different, previously unseen CWEs. Finally, to assess the models' general-purpose code generation ability, we use the standard HumanEval benchmark. Due to space constraints, we provide the comprehensive training configurations in Appendix E.

**Baselines**  To contextualize the performance of our proposed TSP approach, we establish a rigorous hierarchy of baseline models. The foundational comparison is against the Base LLMs—the original, pre-trained foundation models without any security-specific fine-tuning. We then consider SFT , which represents the standard methodology for domain adaptation by fine-tuning on curated datasets of secure code. As a state-of-the-art baseline, we include SafeCoder, a model series specifically engineered for code security. For CodeLlama-7B, we use the officially released SafeCoder model; to ensure a fair comparison for the Qwen2.5-Coder models, we prepare equivalent baselines by fine-tuning them on the same SafeCoder dataset. Crucially, as a critical ablation study for our TSP method, we introduce a Self-Play Fine-Tuning baseline using a self-play mechanism but, importantly, without the structured, tree-based generation of vulnerability nodes that defines our approach.

**Evaluation Methods**  Our evaluation protocol employs a multi-faceted approach, combining static analysis, LLM-based assessment, and general capability benchmarks to ensure a robust analysis. For Python security testing, we utilize CodeQL, a state-of-the-art static analysis (SAST) engine. The primary metric reported is the Security Pass Rate (SPR@1), defined as the percentage of top-1 generated code

*Table 2.* Performance on Python SecurityEval (SPR@1) and HumanEval (pass@k). We compare our TSP method against baselines.

| METHOD | SPR@1 | PASS@1 | PASS@10 |
|---|---|---|---|
| *CodeLlama-7B* | | | |
| BASE LLM | 55.0 | **34.5** | **55.1** |
| SFT | 57.0 | 34.1 | 54.8 |
| SAFECODER | 73.7 | 33.9 | 52.5 |
| SELF-PLAY | 69.6 | 33.3 | 44.9 |
| **TSP (OURS)** | **75.8** | 34.0 | 54.7 |

snippets that pass all relevant security checks. Due to the complexities of C/C++ compilation and environment setup at scale, we employ a highly capable LLM as a security evaluator for these languages. To ensure consistent and reproducible judgments, the evaluator's sampling temperature is fixed at a low value of $\tau = 0.2$. The key metric is the Total Vulnerabilities detected across the test set, where a lower count signifies superior performance. To measure the impact of security fine-tuning on core programming logic, we evaluate all models on the HumanEval benchmark, reporting the standard pass@1 and pass@10 metrics to quantify any potential degradation in general coding ability.

### 4.2. RQ1: Security Performance Enhancement

To answer RQ1, we evaluated the performance of TSP-enhanced models against the established baselines on both language-specific security benchmarks and a general-purpose coding benchmark. The objective was to quantify the direct security uplift provided by our method while also monitoring its impact on core programming capabilities.

The results on the Python benchmarks, presented in Table 2, reveal a clear advantage for our TSP method. For CodeLlama-7B, TSP achieves the highest Security Pass Rate (SPR@1) of 75.8%, surpassing all baselines. This gap between TSP (75.8%) and the Self-Play ablation (69.6%) empirically validates the necessity of our structured vulnerability tree generation. This trend generalizes across Qwen2.5-Coder-7B and Qwen2.5-Coder-3B, where TSP consistently yields the top security performance. Crucially, the HumanEval results show that these significant security gains are achieved with only a minimal and often negligible impact on the models' general-purpose Python coding abilities, demonstrating that TSP does not suffer from catastrophic forgetting.

This trend is strongly corroborated by our C/C++ experiments, shown in Figure 2. For CodeLlama-7B, TSP reduces the vulnerability count to just 94, outperforming all baselines, including the Self-Play approach (103 vulnerabilities) and Base LLM (115). The same winning pattern holds

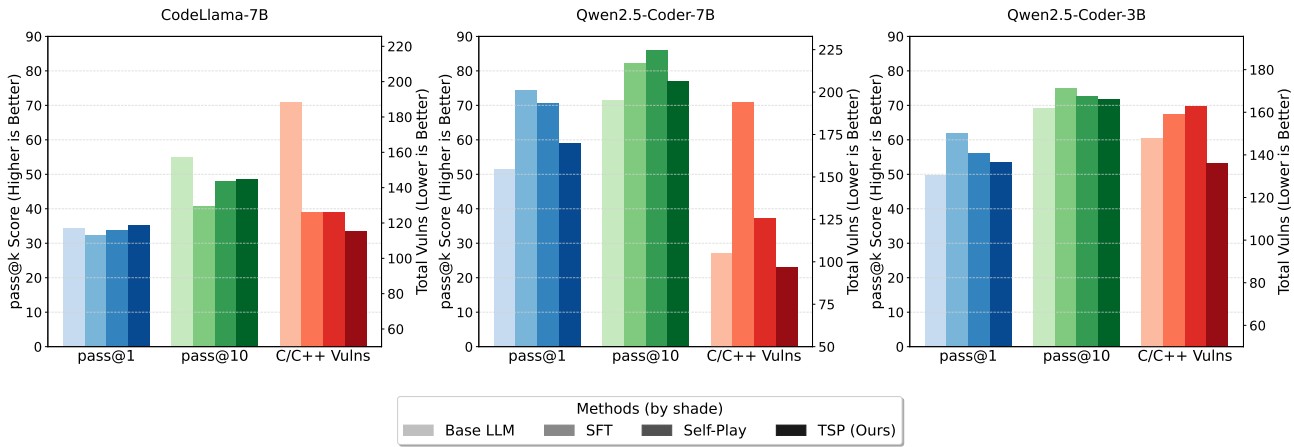

*Figure 2.* Performance comparison of fine-tuning methods on code generation and security tasks. Each subplot corresponds to a specific base model. Within each subplot, we evaluate four methods on the Python HumanEval benchmark (`pass@1` and `pass@10`) and the C/C++ DiverseVul benchmark (`Total Vulns`). For `pass@k` metrics, higher scores indicate better performance, while for `Total Vulns`, lower is better. We distinguish different models by the color shade.

*Table 3.* Language Generalization and HumanEval Performance. Lower is better for Vulns; higher is better for pass@k.

| METHOD | SECURITY VULNS | HUMANEVAL PASS@1 | PASS@10 |
|---|---|---|---|
| *CodeLlama-7B* | | | |
| BASE LLM | 115 | 34.5 | 55.1 |
| SFT | 110 | 32.3 | 40.9 |
| SELF-PLAY | 103 | 33.8 | 48.0 |
| **TSP (OURS)** | **94** | **35.2** | **48.6** |
| *Qwen2.5-Coder-7B* | | | |
| BASE LLM | 106 | 51.6 | 71.5 |
| SFT | 140 | 74.3 | 82.2 |
| SELF-PLAY | 142 | 70.6 | 85.9 |
| **TSP (OURS)** | **57** | **58.9** | **76.9** |
| *Qwen2.5-Coder-3B* | | | |
| BASE LLM | 93 | 49.8 | 69.2 |
| SFT | 107 | 62.0 | 75.0 |
| SELF-PLAY | 130 | 56.0 | 72.8 |
| **TSP (OURS)** | **41** | **53.4** | **71.7** |

*Table 4.* Generalization to Unseen CWEs and Impact on HumanEval Performance. Lower is better for vulnerability counts; higher is better for pass@k.

| METHOD | UNSEEN CWEs FILES | VULNS | HUMANEVAL PASS@1 | PASS@10 |
|---|---|---|---|---|
| *CodeLlama-7B* | | | | |
| BASE LLM | 51 | 64 | **34.5** | **55.1** |
| SFT | 44 | 50 | 32.1 | 54.8 |
| SELF-PLAY | 32 | 42 | 33.3 | 45.2 |
| **TSP (OURS)** | **27** | **34** | 34.3 | 54.7 |
| *Qwen2.5-Coder-7B* | | | | |
| BASE LLM | 26 | 30 | 51.6 | 71.5 |
| SFT | 39 | 58 | **73.3** | 81.9 |
| SELF-PLAY | 34 | 41 | 70.0 | **86.4** |
| **TSP (OURS)** | **21** | **26** | 55.0 | 72.5 |
| *Qwen2.5-Coder-3B* | | | | |
| BASE LLM | 29 | 37 | 49.8 | 69.2 |
| SFT | 46 | 56 | **61.2** | **75.4** |
| SELF-PLAY | 40 | 46 | 55.7 | 72.4 |
| **TSP (OURS)** | **27** | **32** | 55.3 | 73.7 |

for the Qwen2.5-Coder models. The consistent security improvements in both Python and C/C++ and across all three base models affirm that TSP significantly enhances code security. The efficacy of our method stems from its on-policy learning loop, which engages the model in a structured process of exploration, targeted failure analysis, and self-correction, proving critical for mastering the subtleties of code security across different programming languages.

### 4.3. RQ2: Generalization Across Programming Languages

A crucial measure of a security enhancement technique is its ability to generalize beyond the programming language seen during training. We designed an experiment to probe the cross-lingual transfer capability of TSP-trained models, comparing them against our full set of baselines and tracking their general-purpose coding performance on HumanEval. To assess cross-lingual transfer, we leveraged models fine-tuned exclusively on C/C++ (consistent with the training setup in the second experiment of RQ1) and evaluated them on a multi-language test set. This test set was constructed by

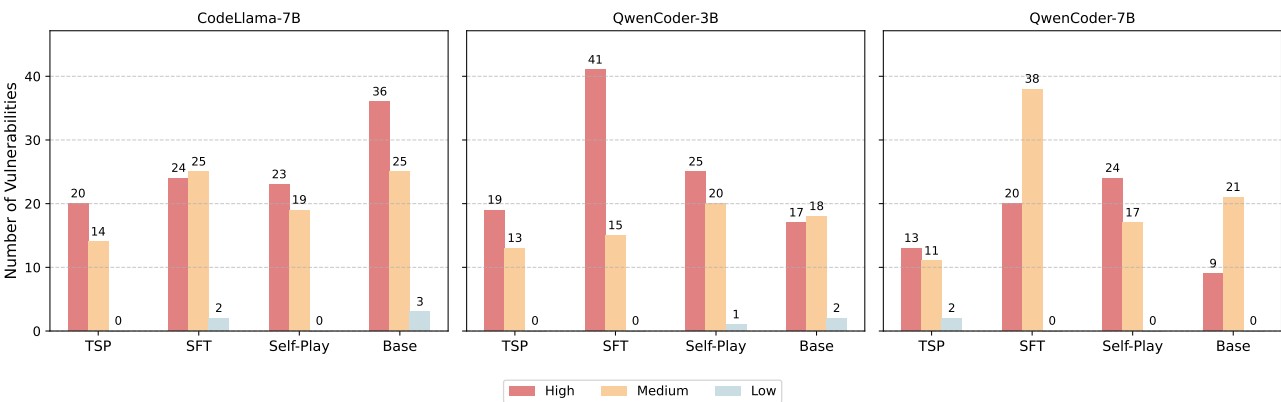

*Figure 3.* Breakdown of vulnerability severity levels (High, Medium, Low) for **unseen CWE types**. The results show that TSP significantly reduces high-severity vulnerabilities compared to baselines (e.g., SFT on Qwen2.5-Coder-7B), demonstrating its effective generalization to novel security threats.

generating description prompts via GPT-4o for code snippets from the public SafeCoder dataset—the same dataset used as the training set in RQ1's first experiment—forming structured prompt-code pairs. The evaluation covered four programming languages: Python, JavaScript, Go, and Ruby, aiming to test whether the learned security principles can generalize across different programming languages.

The results in Table 3 further confirm TSP's strong cross-lingual generalization. For all models, TSP achieves the lowest number of vulnerabilities across all four target languages, despite being trained solely on C/C++ security data. This indicates that TSP helps the model learn language-agnostic security heuristics that transcend specific syntax differences. The HumanEval results provide crucial context: the models' fundamental ability to generate correct code in these languages remains stable, while their capacity to generate secure code is significantly improved. Together, these results provide a compelling answer to RQ2: TSP builds a robust security awareness that generalizes across diverse programming languages.

### 4.4. RQ3: Generalization to Unseen CWE Types

Another critical measure of a security enhancement technique is its ability to generalize beyond the specific vulnerability types seen during training. We designed an experiment to probe this capability in TSP-trained models, focusing on their performance on previously unseen CWE types.

In this experiment, we fine-tuned our models on a C/C++ dataset covering 110 distinct CWE types and then tested them on all 40 CWE types in CodeQL's standard benchmark. This setup measures the ability to learn abstract security principles that can be applied to novel vulnerabilities rather than merely memorizing fixes for known ones. CodeQL's standard benchmark includes a curated set of 40 CWE types that are widely recognized as critical for code security across

programming languages.

The results in Table 4 demonstrate TSP's superior generalization capabilities. TSP consistently achieves the lowest vulnerability counts across all models. For instance, on CodeLlama-7B, TSP reduces the vulnerability count to 27, which is significantly lower than the 50 vulnerabilities detected in the SFT baseline. A similar trend is observed on Qwen2.5-Coder-7B, where TSP achieves 26 vulnerabilities. Notably, SFT exhibits severe overfitting on this model, where the vulnerability count drastically increases from 30 (Base LLM) to 58, whereas TSP effectively mitigates these risks. This suggests that while SFT tends to memorize specific fixes, TSP forces the model to internalize abstract security principles via adversarial self-play, enabling robust defense against previously unencountered vulnerability categories.

Crucially, the severity breakdown in Figure 3 reveals that TSP is particularly effective at mitigating *High-severity* risks. While the SFT model on Qwen2.5-Coder-7B retains a large number of high-severity vulnerabilities, TSP reduces this number to 13. This qualitative improvement suggests that TSP forces the model to internalize abstract security principles via adversarial self-play, enabling it to robustly defend against critical, previously unencountered vulnerabilities where simple memorization fails.

## 5. Limitations

Despite the consistent security improvements demonstrated by TSP, our approach has several limitations that merit discussion. Our CWE-level attribution analysis (Appendix D) reveals a clear pattern: TSP excels at vulnerabilities with local, explicit control flows—such as CWE-215, CWE-079, and CWE-252—where the security-critical decision and its manifestation are co-located, allowing the value network

to prune dangerous branches early. However, TSP underperforms on complex memory and implicit data-flow vulnerabilities, such as CWE-690, CWE-125, and CWE-416, where the unsafe decision and the eventual manifestation are separated by long execution distances with seemingly legitimate intermediate steps, causing the value estimator to misjudge intermediate safety.

More broadly, the current risk node abstraction operates through a primarily token-level lens. While the annotator LLM performs semantic reasoning to contextualize these nodes, many real-world vulnerabilities arise from multi-line data-flow dependencies and cross-variable invariants that cannot be fully captured by node-level annotations. The self-play negative samples also become less challenging as the model improves, potentially limiting the discovery of deeper vulnerability patterns.

Finally, all experiments are conducted on 3B–7B models due to computational constraints. Although we observe no diminishing returns from 3B to 7B, and Pearce et al. (2025) also suggests that even much larger models remain vulnerable, the generalizability of TSP to frontier-scale code LLMs remains an open question.

## 6. Conclusion

In conclusion, this work introduces Tree-like Self-Play, a novel training framework that significantly enhances the security of code generated by Large Language Models. By reframing the learning process around granular, localized decision points corresponding to potential vulnerabilities, TSP provides a highly efficient and targeted training signal. Through a structured, self-play mechanism, our method achieves substantial security improvements that generalize across unseen vulnerabilities and programming languages, demonstrating a clear path toward autonomous model refinement without the need for extensive preference annotation. By focusing on correcting the small mistakes that lead to large failures, TSP provides a powerful and scalable paradigm for building more secure and reliable code generation models.

## Acknowledgments

We gratefully acknowledge the support from Hengheng Zhang as project leader. We also thank the anonymous reviewers for their insightful comments during the rebuttal stage that helped improve this paper.

## Impact Statement

This paper introduces Tree-like Self-Play (TSP), a framework that enhances the security of code generated by Large Language Models (LLMs) by reframing the process as a fine-grained sequential decision task. By training models to identify and correct localized errors at critical decision nodes, our work significantly reduces the propagation of endemic vulnerabilities in software development. The primary societal impact is the advancement of automated secure coding, which improves the reliability of software across multiple programming languages and reduces risks such as SQL injections or buffer overflows. While this research provides a scalable path toward autonomous model refinement, we acknowledge that the effectiveness of the framework is contingent on the quality of risk node identification and requires ongoing monitoring to ensure that optimizing for security does not negatively impact general coding functionality.

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

# A. Related Works

## A.1. Secure Code Generation

LLMs show remarkable progress in code generation, yet the prevalence of security vulnerabilities in their outputs poses a potential security risk. Studies (Jamdade & Liu, 2024; Siddiq & Santos, 2022; Pearce et al., 2025; Perry et al., 2023) indicate that a significant fraction of LLM-generated code, including that models like GPT-3.5, contains security flaws. For instance, up to 40% of GitHub Copilot's code snippets exhibit potential vulnerabilities. Early mitigation efforts include SVEN (He & Vechev, 2023), which steers generation towards security or adversarial testing via training prefixes while maintaining functionality, and SafeCoder (He et al., 2024), which fine-tunes models with security-focused datasets and instruction tuning. However, these approaches are hampered by scarce security-related training data and limitations in their training paradigms.

Further works explore alternative security enhancement strategies. HexaCoder (Hajipour et al., 2024) synthesizes training data with secure oracles and employs a two-step generation method to improve security. ProSec (Xu et al., 2024) proactively enhances LLMs' secure code generation by synthesizing error-prone coding scenarios and applying preference learning. Nevertheless, these methods rely on explicit positive and negative training pair construction, leading to high annotation costs and inefficient training.

Recent works have also advanced secure code generation from complementary angles: AutoSafeCoder (Nunez et al., 2024) integrates static analysis feedback at inference time, and some works (Nie et al., 2025; Wang et al., 2025a) focus on post-hoc vulnerability repair.

## A.2. Post-training for Safe CodeLLMs

Post-training techniques, such as SFT (Ouyang et al., 2022; Li et al., 2024; Xiao et al., 2024), Direct Preference Optimization (DPO) (Rafailov et al., 2023), and Reinforcement Learning from Human Feedback (RLHF) (Christiano et al., 2017; Ouyang et al., 2022; Bai et al., 2022), are pivotal for aligning code generation models with safety objectives. While SFT adapts models to domain-specific tasks through labeled demonstrations, DPO and RLHF further refine outputs by incorporating human preferences or reward signals. However, the effectiveness of these methods heavily relies on the quality and scale of training data.

Existing code vulnerability datasets in the post-training stage often suffer from limited diversity and inconsistent labeling. For instance, widely-used datasets (Fan et al., 2020; Wartschinski et al., 2022) are constructed by extracting vulnerability fix commits, where pre-commit functions are labeled as vulnerable and post-commit versions as secure. However, as revealed in recent studies (He & Vechev, 2023; Croft et al., 2023), this approach introduces noisy security labels, as many code changes in commits (e.g., refactoring or feature updates) are unrelated to security fixes. While SVEN (He & Vechev, 2023) addresses this issue through costly manual inspection to curate high-quality labels, and other works (He et al., 2024; Hajipour et al., 2024; Xu et al., 2024) leverage automated processes and AI annotation to build secure code datasets, their methods remain poor scalable and resource intensive.

TDPO (Zeng et al., 2024) extends DPO to the token level for finer-grained credit assignment, but operates on statically constructed preference pairs. TSP differs by concentrating the learning signal at semantically identified risk nodes through on-policy self-play, generating contextually challenging negative examples that dynamically adapt to the model's evolving failure patterns.

## A.3. Self-play Training Paradigm

Self-play (Heinrich et al., 2015; Tesauro, 1994) trains AI agents through iterative competition with themselves. While pivotal in systems like AlphaGo Zero (Silver et al., 2016) and OpenAI Five (Berner et al., 2019), its principles now shape modern AI alignment as researchers are reframing RLHF as a game-theoretic process (Swamy et al., 2024; Chen et al., 2024; Rosset et al., 2024). For instance, SPIN (Chen et al., 2024) bootstraps a supervised fine-tuned model into a self-improving loop: at each iteration, the model generates responses to compete with its prior version, then learns to discern these from human-crafted examples. This creates a data-efficient curriculum (Li et al., 2024; Xiao et al., 2024) where the model evolves autonomously, mitigating the gap between weak and strong LLMs without additional outer supervision.

Compared to previous research that uses more advanced models or manual annotation during the training of target models to generate better data (Ouyang et al., 2022; Bai et al., 2022; Song et al., 2024; He & Vechev, 2023; He et al., 2024), we directly generate synthetic data from the target model itself through self-play. Not only improves the data utilization efficiency but

also stands out by eliminating the need for additional positive and negative code annotation (which refer to preference pairs), which is a necessary requirement for other training methods (He & Vechev, 2023; He et al., 2024; Hajipour et al., 2024; Xu et al., 2024).

## B. Discussion

Our experimental results robustly demonstrate that TSP offers a significant advancement in training Large Language Models for secure code generation. The consistent outperformance of TSP against strong baselines, including vanilla SFT and standard self-play, is not a mere incremental improvement but points to a fundamental advantage of our approach: the ability to provide granular, context-aware feedback directly at the points where security vulnerabilities are most likely to originate. A key insight from our findings is that global, sequence-level rewards are insufficient for the nuanced task of code security. Vulnerabilities often hinge on single-token errors, such as using an unsafe function or an off-by-one error in a loop condition. TSP's strength lies in its explicit modeling of these critical decision points as nodes in a generation tree. By forcing the model to confront and differentiate between secure and insecure choices at these specific nodes, the learning signal is far more targeted and efficient. The ablation study, where the non-tree-based self-play method consistently underperforms TSP, empirically validates the necessity of this structured, vulnerability-aware approach.

The generalization capabilities of TSP-trained models are particularly noteworthy, suggesting the model is not merely memorizing patches for specific, known vulnerabilities but is instead learning higher-level, abstract principles of secure coding. The ability to mitigate unseen CWEs and to transfer security knowledge from C/C++ to Python indicates that the adversarial nature of the self-play compels the model to develop more robust and generalizable internal representations of security concepts. Nevertheless, our approach is not without limitations. The efficacy of TSP is contingent on the quality of the initial CWE Risk Node annotation. Complex or novel vulnerabilities that do not map cleanly to known syntactic patterns could be missed. Furthermore, while our results showed minimal impact on general coding ability as measured by HumanEval, the potential for alignment taxes—where optimizing for security might subtly degrade functionality—remains a concern that warrants continuous monitoring.

These considerations naturally lead to several promising avenues for future research. A primary direction is to move beyond static, pre-annotated risk nodes and develop methods for their dynamic identification during the self-play process. This would reduce the dependency on initial data curation and empower the model to discover novel or less obvious vulnerability patterns autonomously. Furthermore, the TSP framework itself is highly generalizable. Future work could adapt this node-based self-play paradigm to improve other critical aspects of code quality, such as optimizing for computational efficiency, reducing complexity, or ensuring adherence to specific API usage guidelines, by defining different types of "mistake nodes." Finally, exploring hybrid models that augment TSP's scalable self-play with sparse, high-quality human feedback could create a powerful synergistic learning cycle, combining the scalability of autonomous training with the accuracy of expert oversight for tackling the most complex and ambiguous security cases.

## C. Dataset Construction and Prompting Details

In this section, we provide supplementary details regarding the data construction methodology described in Section 4, including the rigorous human validation criteria and the specific prompts employed.

### C.1. Human Annotation Guidelines

To calculate the inter-rater reliability reported in Section 4.3, human security experts were provided with strict validation criteria. Experts were instructed to label a "CWE Risk Node" as valid only if it met the following conditions:

1. **Causality**: The annotated line must be the exact location where a decision (e.g., a function call or a check) determines the security of the subsequent code path.

2. **Replicability**: An insecure variation of the code at this specific node must plausibly lead to the specified CWE type.

3. **Specificity**: In cases of multi-line logic, only the most decisive line (e.g., the 'if' condition rather than the error handling block) should be annotated.

## C.2. Prompt Templates

We employ a structured prompting strategy to automate the identification of risk nodes. Figure 4 illustrates the exact instructions provided to the LLM.

---

**Prompt Structure for CWE Risk Node Annotation**

**System Prompt Summary:**

- **Role**: Assume the persona of a Code Security Analysis Expert.

- **Task**: Analyze a secure code snippet to identify locations (CWE Risk Nodes) where vulnerabilities could be introduced in an insecure counterpart.

- **Constraints & Format**: For each identified node, output in a strict, structured format:

```
[Node_ID] <n>
[Code Line] <line of code>
[CWE_ID] CWE-<number>
[Description] <Vulnerability description...>
```

- **Criticality Principle**: If multiple lines are susceptible to the same CWE, highlight only the single most critical line.

- - - - -

**User Prompt Template:**

```
Description: {description}
Code:
{code}
```

---

*Figure 4.* The structured prompt template used for automated CWE Risk Node annotation.

## D. CWE-Level Performance Analysis

To provide a granular understanding of TSP's varying effectiveness across vulnerability types, we present a CWE-level breakdown comparing TSP against the standard Self-Play baseline on the C/C++ DiverseVul test set. Table 5 reports the number of detected vulnerabilities per CWE type for both methods.

The results reveal a clear pattern tied to the locality of vulnerability semantics. TSP excels at handling local features and explicit control flows. For instance, CWE-215 (Sensitive Info in Debug Mode) and CWE-079 (XSS) involve immediate semantic violations—an unsafe configuration enabled or a missing input sanitization—that TSP's node-level evaluation readily detects. The value network assigns low rewards at these nodes and prunes dangerous branches early in generation, leading to substantial vulnerability reductions ($-34$ and $-20$, respectively). Similarly, CWE-252 (Unchecked Return Value) and CWE-119 (Buffer Overflow) involve explicit local check omissions at the exact line of the risky operation, making them amenable to TSP's localized learning signal.

Conversely, TSP underperforms on complex memory and implicit data-flow vulnerabilities. In CWE-416 (Use After Free), the variable allocation and the final dangerous invocation are separated by long execution distances, and intermediate steps appear perfectly legitimate. In CWE-690 (NULL Pointer Dereference), the unchecked return value that leads to a NULL pointer and the eventual dereference may be separated by conditional logic spanning dozens of lines. In CWE-457 (Use of Uninitialized Variable), the initialization failure and its downstream exploitation may occur in entirely different control-flow branches. In all these cases, TSP's value estimator suffers from short-sightedness: lacking cross-procedural taint-tracking, the model misjudges intermediate safety and fails to foresee the terminal state collapse.

*Table 5.* **CWE-Level Vulnerability Comparison: TSP vs. Self-Play.** We report the number of detected vulnerabilities on the C/C++ DiverseVul test set. CWE types are grouped by whether TSP outperforms or underperforms the Self-Play baseline.

| Category | CWE ID & Description | TSP | Self-Play | $\Delta$ |
|---|---|---|---|---|
| *TSP Outperforms Self-Play* | | | | |
| Local Config | CWE-215: Sensitive Info in Debug Mode | 4 | 38 | −34 |
| Input Validation | CWE-079: Cross-Site Scripting (XSS) | 16 | 36 | −20 |
| Error Handling | CWE-252: Unchecked Return Value | 0 | 12 | −12 |
| Memory Safety | CWE-119: Buffer Overflow | 20 | 32 | −12 |
| Control Flow | CWE-674: Uncontrolled Recursion | 0 | 4 | −4 |
| *TSP Underperforms Self-Play* | | | | |
| Pointer Safety | CWE-690: NULL Pointer Dereference | 14 | 7 | +7 |
| Memory Safety | CWE-125: Out-of-bounds Read | 16 | 11 | +5 |
| Memory Safety | CWE-416: Use After Free (UAF) | 7 | 3 | +4 |
| Path Traversal | CWE-073: External Control of File Path | 4 | 0 | +4 |
| Initialization | CWE-457: Use of Uninitialized Variable | 15 | 12 | +3 |

# E. Implementation Details

In this section, we provide the detailed hyperparameters and settings used for our proposed Tree-like Self-Play training stage and the subsequent inference phases. We implemented the training pipeline based on the PyTorch framework. For efficient high-throughput inference, we utilized the vLLM library with Tensor Parallelism.

## E.1. TSP Training Hyper-parameters

The TSP alignment stage was optimized using the hyperparameters detailed in Table 6. To accommodate the complex branching structures of the self-play trees (consisting of both golden paths and perturbation branches), we significantly extended the context length to 8192 tokens. We also employed a conservative learning rate of 1e-5 and DeepSpeed ZeRO-2 optimization to ensure training stability and memory efficiency.

*Table 6.* **Hyper-parameters for TSP Alignment Stage.**

| Hyper-parameter | Value |
|---|---|
| Optimizer | AdamW (Fused) |
| Mixed Precision | BF16 |
| Learning Rate Scheduler | Cosine |
| ZeRO Stage (DeepSpeed) | 2 |
| Training Epochs | 3.0 |
| Global Batch Size | 128 |
| Gradient Accumulation Steps | 16 |
| Per-Device Batch Size | 1 |
| Initial Learning Rate | 1.0e-5 |
| Warmup Ratio | 0.1 |
| Model Max Length (Cutoff) | 8192 |
| Dataset Size | 5000 |

## E.2. Inference Hyper-parameters

We employed distinct sampling strategies for the training data generation (Self-Play) and the final model evaluation.

For the Self-Play Generation (where the Opponent Player generates perturbations), we used a higher temperature to ensure diversity in the negative samples.

For Final Evaluation (Main Player), we prioritized deterministic and precise code generation. As detailed in Table 7, we utilized the vLLM engine with a Tensor Parallelism size of 4 to accelerate inference. The temperature was set to 0 (Greedy Decoding) to maximize reproducibility.

*Table 7.* **Hyper-parameters for Inference (Self-Play Generation & Final Evaluation).**

| Hyper-parameter | Value |
|---|---|
| *Self-Play Generation (Training Phase)* | |
| Inference Engine | PyTorch |
| Temperature | 1.0 |
| Top-p | 0.95 |
| Samples per Node | 4 |
| Max New Tokens | 1024 |
| *Final Evaluation (Testing Phase)* | |
| Inference Engine | vLLM |
| Tensor Parallelism (TP) | 4 |
| Temperature | 0 (Greedy) |
| Top-p | 0.95 |
| Max New Tokens | 4096 |
| GPU Memory Utilization | 0.85 |

### E.3. Baseline Hyper-parameters

To ensure a fair comparison, we utilized standard academic hyper-parameter settings for the baseline methods. Table 8 and Table 9 provide the configurations used for the Baseline SFT and Baseline DPO (Self-Play with standard preference pairs) experiments, respectively.

*Table 8.* **Hyper-parameters for Baseline SFT Training.**

| Hyper-parameter | Value |
|---|---|
| Optimizer | AdamW |
| Learning Rate | 2e-5 |
| LR Scheduler | Cosine |
| Warmup Ratio | 0.03 |
| Weight Decay | 0.0 |
| Training Epochs | 3 |
| Global Batch Size | 128 |
| Gradient Accumulation | 8 |
| Per-Device Batch Size | 2 |
| Mixed Precision | BF16 |
| Model Max Length | 2048 |
| DeepSpeed Stage | ZeRO-3 |

## F. Case Study: Real-world Cryptographic API Scenario

To clearly and intuitively illustrate the operational mechanism and effectiveness of our proposed TSP framework, we present a concrete case study grounded in a real-world cryptographic API scenario. This example focuses on using OpenSSL's Cryptographic Message Syntax (CMS) API to decode PKCS7 data, demonstrating how a failure to properly validate the return value of a critical CMS function can open the door to a padding-oracle vulnerability.

*Table 9.* **Hyper-parameters for Baseline DPO Training.**

| Hyper-parameter | Value |
|---|---|
| Optimizer | AdamW |
| Learning Rate | 1e-5 |
| LR Scheduler | Cosine |
| Warmup Ratio | 0.1 |
| KL Penalty ($\beta$) | 0.1 |
| Training Epochs | 3 |
| Global Batch Size | 64 |
| Gradient Accumulation | 8 |
| Per-Device Batch Size | 1 |
| Mixed Precision | BF16 |
| Model Max Length | 4096 |
| DeepSpeed Stage | ZeRO-2 |

### F.1. Scenario Overview

Initially, the model is given the following input prompt:

> **User Prompt**
>
> Implement a C function using OpenSSL's CMS API to securely decode PKCS7 data by validating the decrypted key length against the cipher's default key length, ensuring protection against padding oracle attacks.

The ground-truth $y$ follows standard OpenSSL practices, as shown in Listing 1. The function `CMS_sign_receipt` performs CMS receipt signing while adhering to essential security validations, including correct key length checks and return value validation of cryptographic primitives.

```c
CMS_ContentInfo *CMS_sign_receipt(CMS_SignerInfo *si,
                                  X509 *signcert, EVP_PKEY *pkey,
                                  STACK_OF(X509) *certs, unsigned int flags) {
    CMS_SignerInfo *rct_si;
    CMS_ContentInfo *cms = NULL;
    ASN1_OCTET_STRING **pos, *os;
    BIO *rct_cont = NULL;
    int r = 0;

    flags &= ~(CMS_STREAM | CMS_TEXT);
    flags |= CMS_PARTIAL | CMS_BINARY | CMS_DETACHED;

    if (!pkey || !signcert) {
        CMSerr(CMS_F_CMS_SIGN_RECEIPT, CMS_R_NO_KEY_OR_CERT);
        return NULL;
    }

    cms = CMS_sign(NULL, NULL, certs, NULL, flags);
    if (!cms) goto err;

    if (!CMS_set1_eContentType(cms, OBJ_nid2obj(NID_id_smime_ct_receipt)))
        goto err;

    rct_si = CMS_add1_signer(cms, signcert, pkey, NULL, flags);
    if (!rct_si) {
        CMSerr(CMS_F_CMS_SIGN_RECEIPT, CMS_R_ADD_SIGNER_ERROR);
        goto err;
    }

    os = cms_encode_Receipt(si);
```

```
31      if (!os) goto err;
32
33      rct_cont = BIO_new_mem_buf(os->data, os->length);
34      if (!rct_cont) goto err;
35
36      if (!cms_msgSigDigest_add1(rct_si, si))
37          goto err;
38
39      // Security-critical node
40      if (!CMS_final(cms, rct_cont, NULL, flags))
41          goto err;
42
43      pos = CMS_get0_content(cms);
44      *pos = os;
45
46      r = 1;
47  err:
48      BIO_free(rct_cont);
49      if (r) return cms;
50      CMS_ContentInfo_free(cms);
51      return NULL;
52  }
```

*Listing 1.* Golden path implementation for PKCS7 decoding using OpenSSL CMS API.

### F.2. Risk Node Identification

Through our automated annotation pipeline, the TSP framework identifies potential vulnerability points within the golden path. In this case, the critical node resides in the call to `CMS_final`:

- **Line**: `if (!CMS_final (cms, rct_cont, NULL, flags)) goto err;`

- **CWE-ID**: CWE-754 (Improper Check for Unusual or Exceptional Conditions).

- **Rationale**: `CMS_final` may fail due to internal cryptographic validation errors. Ignoring its return value can lead to the use of an incompletely initialized CMS structure, resulting in unstable behavior or exposure to padding oracle attacks.

### F.3. Tree-like Self-Play Dynamics

At this identified node, TSP models the generation process as a branching decision:

- **Golden Path Branch** ($y$): The model generates the secure continuation by validating the return value:

```
if (!CMS_final(cms, rct_cont, NULL, flags))
    goto err;
```

- **Perturbation Branch** ($y'_v$): An adversarial player generates a superficially plausible yet unsafe variant where the return value is ignored:

```
// Unsafe alternative: return value ignored
CMS_final(cms, rct_cont, NULL, flags); // Vulnerable
```

During training, the TSP framework treats $y'_v$ as a structured in-context negative example, applying contrastive learning to bias the model toward secure continuations and penalize high-risk behaviors. Code paths outside the golden path, with their hidden vulnerabilities and irregular implementations, are gradually exposed through continuous negative commonsense feedback, prompting the model to further improve the security and compliance of its generated code.

