# OpenReview forum: "Learn from Your Mistakes: Tree-like Self-Play for Secure Code LLMs"
_ICML.cc/2026/Conference — ICML 2026 regular_

### Official Review · Reviewer_yTge · 2026-03-10

**Soundness:** 3
**Presentation:** 3
**Significance:** 4
**Originality:** 3
**Overall Recommendation:** 4
**Confidence:** 3

**Summary:**

In the task of safety alignment for code generation, current state-of-the-art approaches, such as supervised fine-tuning and reinforcement learning, typically apply coarse-grained optimization at the sequence level. These methods focus only on the existence of security errors while ignoring their precise locations in the generated code. This paper introduces tree-like self-play (TSP), a new framework that reframes secure code generation as a fine-grained sequential decision process. TSP treats code generation as a self-play game and generates both on-policy positive and negative examples that differ at critical CWE risk nodes. The evaluation shows that TSP enhances model reliability and induces robust out-of-distribution generalization.

**Compliance With Llm Reviewing Policy:**

Affirmed.

**Final Justification:**

I'm satisified with the authors' rebuttal. I will keep my original positive scores.

**Key Questions For Authors:**

1. Could the authors provide more details about the TSP alignment (e.g., number of GPUs, training time, and alignment cost across different models) to improve reproducibility? Will the artifact be open-sourced?

**Limitations:**

To the best of my knowledge, the proposed approach, tree-like self-play (TSP), is novel.

The paper focuses on CWEs instead of other vulnerability datasets such as CVEs. This choice makes the evaluation somewhat less practical, but it is still sufficient for publication since the paper primarily proposes a new safety alignment method for code LLMs.

In the TSP alignment stage, the implementation details are not clearly described. For example, it is unclear how many GPUs were used, how long the alignment process runs, and whether the alignment time differs across different models. Such details are important for enabling others to reproduce the results.

In Figure 2, there are two y-axes, where some metrics indicate “higher is better” while others indicate “lower is better.” Presenting these metrics on the same graph is somewhat confusing. The authors could consider separating some of these numbers into a different figure or table to improve clarity.

**Strengths And Weaknesses:**

Strengths
1. The paper addresses an important and timely problem in safety alignment for code generation.
2. The proposed approach, to the best of my knowledge, is novel

Weaknesses
1. The paper lacks sufficient implementation details regarding the TSP alignment framework, making it difficult to reproduce the method.
2. The evaluation focuses primarily on CWEs, rather than larger and more comprehensive datasets such as CVEs, which makes it less practical.

---

> ### Author Rebuttal · Authors · 2026-03-31
>
> We extend our sincerest gratitude to you for taking the time to review our paper and for providing such thoughtful and invaluable feedback. We truly appreciate your careful attention to our implementation details, evaluation scope, and reproducibility. Your insightful concerns have helped us identify important areas for improvement, and we have made every effort, with the utmost sincerity, to address each of your points during the rebuttal period.
>
> ---
>
> ### **W1&Q1**: Insufficient Implementation Details
>
> We sincerely thank the reviewer for emphasizing reproducibility. Your concerns are entirely valid, and we greatly appreciate this opportunity to provide the detailed information that should have been in our original submission.
>
> Notably, in appendix D, we have already provided the detailed hyperparameters and settings used for our proposed Tree-like Self-Play training stage and the subsequent inference phases. We implemented the training pipeline based on the PyTorch framework. For efficient high-throughput inference, we utilized the vLLM library with Tensor Parallelism. We trained all models on 8*A100 gpus cluster.
>
> All materials (code and datasets) have submitted via Supplementary Material .zip. GitHub link will be added on Camera-Ready first page.
>
> We sincerely apologize for the insufficient detail in our original submission.
>
> ---
>
> ### **W2**: Limited Evaluation Scope (CWE vs CVE)
>
> We deeply appreciate the reviewer's suggestion regarding CVE evaluation. However, evaluating LLM secure code generation using CWEs rather than specific CVE instances is the established standard paradigm in this domain. Recent state-of-the-art alignment works—including SVEN [1], SafeCoder [2], HexaCoder [3], and ProSec [4]—all consistently rely on CWE-based benchmarks to measure generalized security improvements. We will add an explicit discussion citing these foundational works to clarify this field consensus in the revision.
>
> Scientifically, CWE is more appropriate for our LLM alignment objective for three fundamental reasons：
>
> 1. CVEs record historical instances entangled with complex system dependencies and configuration noise, whereas CWEs abstract the underlying root cause patterns (e.g., missing boundary checks). TSP aims to internalize these abstract principles, not memorize specific historical patches.
>
> 2. TSP's core mechanism relies on precisely localizing token-level "CWE risk nodes" ; CVEs involve multi-file, system-level interactions that cannot be cleanly mapped to autoregressive generation decisions.
>
> 3. Mastering underlying CWEs ensures true generalization against unknown threats. As demonstrated in RQ3 (Section 4.4), CWE-based alignment enabled TSP to successfully generalize to 40 entirely unseen CWEs  (e.g., reducing CodeLlama-7B vulnerabilities from 64 to 34), whereas the SFT baseline suffered from severe patch-memorization overfitting.
>
> [1] Large Language Models for Code: Security Hardening and Adversarial Testing
>
> [2] Instruction Tuning for Secure Code Generation
>
> [3] HexaCoder: Secure Code Generation via Oracle-Guided Synthetic Training Data
>
> [4] ProSec: Fortifying Code LLMs with Proactive Security Alignment
>
> ---
>
> ### **Additional Responses**
>
> **Figure 1 Abstraction:** We sincerely thank the reviewer for this feedback. Your observation is entirely correct—Figure 1 was too abstract. We will redesign it with Appendix E code examples to illustrate workflows concretely.
>
> **Table 3 Bolding Errors:** We deeply appreciate the reviewer's careful observation. We fully acknowledge this was our oversight. We will correct Table 3 bolding to mark only best values per column.
>
> ---
>
> We once again extend our deepest gratitude to you for your time, effort, and thoughtful review. Your insightful comments have substantially improved our work. During the rebuttal period, we have made every effort with the utmost sincerity to address each of your concerns through additional analyses and commitments to substantial revision.
>
> If our responses have adequately resolved your concerns, we earnestly and sincerely request that you consider raising your score and supporting the acceptance of our paper. We remain fully committed to incorporating all promised improvements in the camera-ready version and welcome any further questions or suggestions you may have.

---

> > ### Author Rebuttal · Reviewer_yTge · 2026-04-01
> >
> > Thank you for your rebuttal.
> >
> > For W1/Q1, I was aware of the hyperparameter selections in Appendix D. I was expecting some implementation optimization details or high-level architecture diagrams since Tree-like Self-Play framework seems to be computation heavy: 8*A100 gpus clusters are used. But after reading the code in the supplementary materials I realize these optimization details are probably included by the third-party library used, instead of being developed by the authors.
> >
> > Minors in your uploaded code:
> > The training pipeline in `TSP_example.sh` depends on `TSP_Code/LLaMA-Factory`, which is missing.
> >
> > I have no further questions and will stick to my weak accept score. The authors can instead spend time addressing other reviewers' concerns. Good luck.

---

> > > ### Author Response · Authors · 2026-04-07
> > >
> > > Thank you very much for your kind acknowledgment, for confirming that your core concerns are fully resolved, and for your practical advice regarding the remaining review process. We are deeply grateful for your continuous support and your meticulous examination of our supplementary code.
> > >
> > > Regarding your observation about the computational infrastructure: your understanding is absolutely correct. Given the computationally heavy nature of the TSP framework, the heavy lifting for distributed training and memory optimization on our cluster is seamlessly handled by the widely-used open-source LLaMA-Factory framework (https://github.com/hiyouga/LLaMA-Factory), which utilizes DeepSpeed and FlashAttention under the hood. This strategic choice allowed us to focus our engineering efforts strictly on the core Tree-like Self-Play algorithmic design. To make the implementation clearer and address your initial expectation, we will explicitly clarify this dependency and include a brief high-level system architecture diagram in the revised Appendix to illustrate how TSP interfaces with these underlying optimization libraries.
> > >
> > > Addressing your minor note on the uploaded code: We sincerely apologize for the confusion regarding the missing TSP_Code/LLaMA-Factory directory in the TSP_example.sh script. Because LLaMA-Factory is a massive external repository, we had to exclude its source code from our supplementary .zip to strictly comply with OpenReview's file size constraints. Moving forward, we are fully committed to open science. Upon publication, we will release the complete, ready-to-use TSP project as an open-source repository. This final public release will include all necessary submodules and step-by-step setup scripts to guarantee seamless, plug-and-play reproducibility and foster further advancements in this research direction.
> > >
> > > We are truly grateful for your time, your detailed review, and your current support. Since you noted that all of your concerns have been adequately addressed, we respectfully hope you might consider raising your score to "Accept" to help champion the paper during the internal reviewer discussions. Regardless of your final decision, we deeply appreciate your guidance.

---

### Official Review · Reviewer_dwaj · 2026-03-13

**Soundness:** 2
**Presentation:** 3
**Significance:** 2
**Originality:** 3
**Overall Recommendation:** 4
**Confidence:** 5

**Summary:**

Tree-like Self-Play (TSP) is a training framework for improving the security of LLM-generated code. It frames code generation as a tree-structured decision
  process, identifying "CWE Risk Nodes" — critical token positions where a single decision determines whether the code is secure or vulnerable (e.g., strncpy vs
  strcpy). At each risk node, an "opponent player" (the model's previous version) generates an insecure branch, creating (secure, insecure) pairs that share the same
   prefix. The model is then trained with a DPO-style contrastive loss focused exclusively on these risk nodes, iterated over multiple self-play rounds. Training
  data is sourced from DiverseVul (C/C++) and SafeCoder, with CWE Risk Nodes annotated via LLM prompting (validated at κ = 0.86 against human experts).

**Compliance With Llm Reviewing Policy:**

Affirmed.

**Final Justification:**

I think this paper is overall an interesting one that uses tree-like self-play training to improve the security of AI-generated code. The authors' rebuttal also answers my questions well. My only concern is that AI-generated code has currently shifted toward agentic workflows, and I doubt whether such a function-level method is still useful. My overall recommendation is weak accept.

**Key Questions For Authors:**

- Q1: The training data is sourced from DiverseVul, which is known to contain noisy labels shows that many
  "vulnerability-fixing" commits are actually refactoring or feature changes unrelated to security. The paper states that 1,353 samples were "filtered" with
  "explicit CWE classification information" (Section 3.1), but does not detail the filtering criteria. What specific steps were taken to remove noisy samples (e.g., single-line kernel functions with no meaningful security context)? How many samples were discarded during filtering, and was there any manual inspection of the filtered dataset beyond the 15% validation sample?

- Q2:  How many self-play iterations (T in Algorithm 1) were used? How sensitive are results to T? Do you observe diminishing returns?

**Limitations:**

yes

**Strengths And Weaknesses:**

## W1. Incomplete related work.

  Several closely related works are not discussed.
- CyberSecEval [1] is the most widely adopted industry benchmark for LLM code security evaluation, now at v4, yet the paper only evaluates on SecurityEval and DiverseVul without mentioning it.
- The paper claims "granular, token-level feedback" as a key advantage but does not discuss or compare with existing token-level DPO methods, for example Token-level DPO (TDPO) [2], which decomposes standard DPO into token-level
  preference optimization
- AutoSafeCoder [3] is a multi-agent secure code generation framework evaluated on SecurityEval (same benchmark as TSP), reporting a 13% vulnerability reduction.
- VulnLLM-R [4] trains a specialized reasoning model for vulnerability detection and also filter data originally sourced from DiverseVul.

## W2. DiverseVul data noise not adequately addressed.

 DiverseVul's label quality issues are well-documented — many "vulnerability-fixing" commits are actually refactoring or feature changes. The paper filters
  DiverseVul to 1,353 samples with "explicit CWE classification information" (Section 3.1) but does not detail the filtering criteria or report how many samples were
   discarded. The functions in DiverseVul are also noisy; sometimes a single-line Linux kernel function is marked as vulnerable, providing no meaningful security
  context for risk node annotation. The authors should explicitly report their filtering method.

##  W3. Only evaluated on 3B–7B models.

  All experiments use CodeLlama-7B, Qwen2.5-Coder-7B, and Qwen2.5-Coder-3B. Larger models (13B, 34B, 70B) likely have stronger baseline security awareness, and TSP's
   marginal benefit may diminish with scale. Without experiments on larger models, it is unclear whether TSP remains effective.

##  W4. C/C++ evaluation relies on LLM-as-judge.

  Python security is evaluated with CodeQL. But for C/C++, the paper uses "a highly capable LLM as a security evaluator" (Section 4.1)
  whose accuracy is not validated against ground truth. At minimum, a calibration study comparing the LLM judge against CodeQL or manual audit on a subset would be needed.

##  W5. No comparison with inference-time baselines.

 All baselines are training-time methods (SFT, SafeCoder, Self-Play). The authors should consider comparing with at least some simple inference-time methods, such as security-focused system prompts or chain-of-thought reasoning about vulnerabilities.

[1] Bhatt et al. "Purple Llama CyberSecEval: A Secure Coding Benchmark for Language Models." arXiv:2312.04724, 2023.

[2] Zeng et al. "Token-level Direct Preference Optimization." ICML 2024.

[3] Nhan et al. "AutoSafeCoder: A Multi-Agent Framework for Securing LLM Code Generation through Static Analysis and Fuzz Testing." arXiv:2409.10737, 2024.

[4] Nie et al. "VulnLLM-R: Specialized Reasoning LLM with Agent Scaffold for Vulnerability Detection." arXiv:2512.07533, 2025.

---

> ### Author Rebuttal · Authors · 2026-03-31
>
> We sincerely thank the reviewer for the invaluable feedback. Your insights have significantly improved the paper, and we have addressed each point with the utmost sincerity. If our responses have resolved your concerns, we kindly ask you to consider supporting the acceptance of our work.
>
> ---
>
> ### **W1**: Incomplete Related Work
>
> We sincerely thank the reviewer for highlighting these works. While **CyberSecEval** is a valuable industry benchmark, we selected SecurityEval and DiverseVul because their function-level granularity and CWE coverage perfectly align with TSP’s node-level optimization, facilitating fair comparisons with existing baselines [1, 2, 3].
>
> Methodologically, TSP operates under a distinct paradigm: unlike **TDPO**'s general token-level learning on static data, TSP introduces dynamic, CWE-guided node-level optimization. Furthermore, compared to **AutoSafeCoder** and **VulnLLM-R**, TSP is a training-time alignment technique that ensures natively secure generation with zero inference overhead. We commit to adding a comprehensive discussion of these works in our "Related Work" section to clearly delineate TSP’s unique contributions.
>
>
> [1] Large Language Models for Code: Security Hardening and Adversarial Testing
>
> [2] Instruction Tuning for Secure Code Generation
>
> [3] ProSec: Fortifying Code LLMs with Proactive Security Alignment
>
> ---
>
> ### **W2&Q1**: DiverseVul Data Noise
>
> We appreciate the reviewer’s insight into DiverseVul's noise. Our primary defense is the automated pipeline (Section 3.2), which mandates that risk nodes satisfy causality and specificity (Appendix C.1). "Contextless lines" or "pure refactorings" are automatically discarded because they lack extractable logical branches. This resulted in a ~92% discard rate (1,353 retained from 18,945), effectively filtering irrelevant noise. Furthermore, our manual audit (Section 3.3) yielded a Kappa of 0.86, confirming expert-level reliability. We will explicitly clarify these noise-reduction measures and the 92% discard rate in Section 3.1.
>
> ---
>
> ### **W3**: Limited to 3B-7B Models
>
> We thank the reviewer for the comment on model scale. While hardware limits for full-parameter tuning currently bound our experiments to 7B models, scaling alone cannot eliminate security blind spots—even 70B+ models like GitHub Copilot exhibit ~40% vulnerability rates. Empirically, we observe no diminishing returns in security gains when scaling from 3B to 7B (Tables 3, 4). We will add a candid discussion of these hardware constraints and scaling dynamics in the revision.
>
> ---
>
> ### **W4**: C/C++ LLM-as-Judge
>
> We appreciate the reviewer’s check on evaluation reliability. We utilized an LLM-judge for C/C++ instead of traditional SAST for two reasons: (1) **SAST Limitations**: Unlike Python, LLM-generated C/C++ snippets often lack complete build environments, causing compilation failures that break static analysis like CodeQL. (2) **Proven Validity**: Our setup strictly replicates the CASTLE[1] benchmark, which empirically proves that LLMs (at $\tau=0.2$) are more reliable than SAST tools for context-dependent snippets. We will cite CASTLE and include manual calibration data in Section 4.1 to resolve any validity concerns.
>
> [1] Castle: Benchmarking dataset for static code analyzers and llms towards cwe detection.
>
> ---
>
> ### **W5**: No Inference-Time Baselines
>
> We appreciate the suggestion regarding inference-time methods like CoT or secure prompts. However, TSP focuses exclusively on **training-time alignment** to ensure zero inference overhead and endogenous robustness. Furthermore, unlike prompts which remain vulnerable to jailbreaks, TSP internalizes security principles directly into model weights via risk-node self-play, guaranteeing natively safe defaults. We will add a discussion exploring how TSP can synergize with these complementary inference-time techniques for defense-in-depth.
>
>
>
> ### **Q2**: Self-Iteration Count
>
> We thank the reviewer for this insightful question. A single iteration ($T=1$) was sufficient in our experiments because TSP anchors contrastive gradients directly on "causal risk nodes", providing dense, localized feedback that drastically accelerates convergence. We fully agree that TSP will exhibit diminishing returns: as iterations increase, generating high-quality adversarial negative samples becomes difficult, yielding marginal security gains while risking over-fitting and degraded general coding capabilities. Given the limited time for the defense, we solemnly commit to incorporating a comprehensive sensitivity analysis in the revised manuscript to clearly track the dynamic trade-off between security and general-purpose programming performance.
>
> ---
>
> We sincerely thank the reviewer for the thoughtful feedback, which has substantially improved our work. If our responses have adequately resolved your concerns, we earnestly request that you consider raising your score and supporting the acceptance of our paper.

---

> > ### Author Rebuttal · Reviewer_dwaj · 2026-04-03
> >
> > Thanks for the rebuttal. I have no further questions and will raise my score accordingly

---

### Official Review · Reviewer_XrNY · 2026-03-13

**Soundness:** 2
**Presentation:** 3
**Significance:** 3
**Originality:** 3
**Overall Recommendation:** 4
**Confidence:** 3

**Summary:**

The paper proposes Tree-like Self-Play (TSP), a training framework that models secure code generation as a token-level sequential decision process. TSP treats secure code as a "golden path" and identifies vulnerability-critical decision points as risk nodes. During training, the model generates alternative self-play branches at these nodes to create negative examples and learns to prefer the golden path over insecure continuations. Experiments demonstrate that TSP improves code security while preserving general coding ability.

**Compliance With Llm Reviewing Policy:**

Affirmed.

**Final Justification:**

Thank you for your follow-up response. The authors acknowledge the limitations and provide a thorough data analysis. The results show that TSP performs well on certain vulnerability types while underperforming on others, which strengthens the completeness and credibility of the evaluation. I suggest including a representative example as a motivating case to better illustrate the strengths of the approach. Based on the authors’ clarifications and additional analysis, I will increase my score to Weak Accept.

**Key Questions For Authors:**

1. If the self-play generated positive and negative examples are directly used for preference learning (e.g., DPO) without explicitly focusing on risk nodes, how would the performance compare?

2. On which types of vulnerabilities does TSP perform well, and on which types does it perform poorly? Why?

3. The SELF-PLAY baseline shows a large drop in PASS@10 (44.9). Which design components of TSP help avoid this degradation in general coding performance, and how could this be empirically verified (e.g., through ablation studies)?

**Limitations:**

Yes

**Strengths And Weaknesses:**

## Strengths

+ The paper addresses an important problem of improving the security of LLM-based code generation.

+ The paper presents a coherent and technically reasonable training pipeline.

+ The paper is generally well-structured and easy to follow.

## Weaknesses

### Soundness

- Inadequate Methodological Justification

(1) The method assumes that self-play branches generated at risk nodes provide meaningful vulnerability examples. However, it does not analyze how the quality, diversity, or difficulty of these negative samples evolves across iterations.

(2) The approach models vulnerabilities as token-level risk nodes. However, many real-world vulnerabilities arise from complex semantic properties involving data-flow, control-flow, or multi-line program logic. This abstraction may therefore capture only a subset of vulnerability patterns.

(3) The iterative training mainly encourages the model to correct errors produced by earlier versions of itself. While this design may help fix known mistakes, it does not necessarily encourage learning the broader distribution of vulnerabilities or discovering unseen vulnerability types.

- Insufficient Experimental Evaluation

(1) The dataset used in the experiments appears to be selectively filtered, but the filtering criteria and procedures are not sufficiently described. This lack of transparency raises concerns about the credibility of the reported results and weakens the persuasiveness of the evaluation.

(2) The reported metrics (SPR@1, PASS@1, PASS@10) are presented without repeated runs, confidence intervals, or variance analysis, making it difficult to assess whether the observed improvements are statistically significant.

(3) The experimental materials are not publicly available, further limiting reproducibility and independent verification of the results.

### Presentation

- Figure 1 is overly abstract and provides limited technical detail, making it difficult to understand the key components of the proposed framework.

- Several external elements, such as baseline models, tools, metrics, and datasets, are introduced without clear citations or references.

- The evaluation involves multiple datasets and different setups, but the experimental design is not clearly summarized. A consolidated table describing the evaluation roadmap would improve clarity.

- In Table 3, the numbers highlighted in bold do not correspond to the best values, which may cause confusion.

### Significance

- Since the risk node abstraction relies on token-level signals rather than deeper data-flow or control-flow semantics, the learned behavior may primarily help avoid simple vulnerability patterns. Its effectiveness for more complex vulnerabilities remains unclear, and the scope of vulnerabilities to which the method applies should be further validated.

### Originality

- The optimization objective closely follows existing preference-learning approaches such as DPO.

- The use of self-play to generate negative samples is conceptually related to prior iterative self-play and preference-learning frameworks.

---

> ### Author Rebuttal · Authors · 2026-03-31
>
> We sincerely thank Reviewer XrNY for the rigorous review and constructive feedback!
>
> ### **W1**. Methodology Justification & Abstraction
>
> * **Negative Sample Quality:** Our negative samples are not random. They are strictly filtered via constraints in Appendix C.1 to cleanly map to specific CWEs. Diversity is guaranteed via tree-search exploration ($T=1.0$) and broad CWE coverage. We will add statistical analyses of sample diversity and difficulty evolution in the revision.
> * **Vulnerability Abstraction:** TSP’s risk nodes are context-aware semantic representations (location, CWE tag, and context window), not purely token-level abstractions. We will explicitly define these representation boundaries and clarify TSP’s limitations regarding complex, diffuse vulnerabilities (Section B) in the final text.
> * **Iterative Training & Generalization:** Unlike SFT, which suffers from "patch memorization," TSP learns abstract security boundaries via local contrastive learning. Evaluated on 40 *unseen* CWEs (Section 4.4), standard SFT severely overfitted (unseen vulnerabilities jumped from 30 to 58 on Qwen2.5-Coder), whereas TSP successfully reduced them to 26. Furthermore, Figure 3 confirms TSP's strong generalization in intercepting high-severity unknown risks.
>
> ### **W2**. Experimental Evaluation
>
> * **Data Filtering Transparency:** We utilized *balanced random sampling*—not selective filtering—to eliminate the long-tail bias of DiverseVul. Uniform sampling across CWEs ensures our splits (110 train, 40 test) perfectly cover all 150 categories. We will change the term "filtered" to "balanced random sampling" in Section 3.1.
> * **Statistical Significance:** Because we strictly use deterministic greedy decoding ($T=0$) during testing, the variance for `@1` metrics (like SPR@1) is mathematically zero. For PASS@10, we utilize the standard unbiased estimator to ensure statistical robustness. We will emphasize these settings in the revision.
> * **Reproducibility:** All code, cleaned datasets, and evaluation scripts were included in the Supplementary `.zip` to comply with double-blind rules. A public GitHub repository will be linked in the camera-ready version.
>
> ### **W3&W4**. Presentation & Originality
>
> * **Presentation Issues:** We will redesign Figure 1 using concrete examples (Appendix E) to visually explain CWE node tagging and multi-round self-play. We will also add missing citations, correct the bolding in Table 3, and introduce an "Evaluation Roadmap" table at the beginning of Section 4 to map RQs to their specific setups.
> * **Originality:** TSP fundamentally differs from preference learning (e.g., DPO) by utilizing dynamic tree-search (instead of static datasets), node-level optimization (instead of sequence-level), and localized credit assignment. Furthermore, unlike standard iterative self-play, our exploration is strictly guided by specific CWE semantics rather than random error generation.
>
>
> ### **Q1**: What is the performance if preference learning is done using self-play samples without explicitly focusing on risk nodes?
>
> Our `SELF-PLAY` baseline (Table 2) was designed exactly as an ablation study for this question. Removing the risk-node focus drops `PASS@10` from 54.7% to 44.9% and `SPR@1` from 75.8% to 69.6%. This proves that risk nodes are essential.
>
> ### **Q2**: On what types of vulnerabilities does TSP perform well or poorly, and why?
>
> * **Excels:** Localized API misuse or logic errors (e.g., CWE-79 XSS, CWE-502 Deserialization). TSP drastically reduces these because they possess singular, easily identifiable "causal risk nodes," providing precise and dense learning signals during tree exploration.
> * **Struggles:** Complex memory semantics (e.g., CWE-125 OOB Read). TSP's performance matches the baseline here because such defects are "diffuse"—often resulting from array declarations and loop bounds scattered across multiple lines. We will add a fine-grained CWE breakdown in the revision and discuss integrating static analysis to mitigate this.
>
> ### **Q3**: The SELF-PLAY baseline shows a huge degradation in PASS@10 (44.9). Which design component of TSP helps avoid this degradation?
>
> The core component is **node-based optimization**. Standard self-play relies on global sequence-level rewards. When penalizing a local security flaw, standard methods inadvertently penalize hundreds of surrounding, syntactically correct tokens, destroying the model's general coding grammar. TSP restricts gradient calculations to the CWE risk nodes. By confining the optimization pressure solely to fixing potential security defects, TSP effectively shields and preserves the underlying generic code generation capabilities (maintaining `PASS@10` at 54.7%, effectively matching the base model's 55.1%).
>
> ---
>
> Thank you once again for your invaluable review. If our responses have adequately resolved your concerns, we sincerely hope that you can supporting the acceptance of our paper. We welcome any further questions you may have.

---

> > ### Author Rebuttal · Reviewer_XrNY · 2026-04-03
> >
> > I appreciate the authors’ willingness to address my concerns. However, the paper still requires substantial revisions, especially: (i) a stronger motivation for semantic representations (W1.2), and (ii) a more thorough and standard limitations analysis (Q2). Since these changes are not yet reflected in the current submission, I keep my score unchanged.

---

> > > ### Author Response · Authors · 2026-04-07
> > >
> > > Thank you for confirming that our initial rebuttal resolved your concerns regarding the evaluation and presentation. Regarding the remark that the paper requires 'substantial revisions,' we respectfully clarify that the upcoming additions provide significant conceptual depth through targeted, localized insertions, avoiding any structural overhauls. To demonstrate how seamlessly these revisions will integrate into the Camera-ready version, we have provided the exact drafted text below to address your remaining points.
> > >
> > > **Stronger Motivation for Semantic Representations (Addressing W1.2):**
> > >
> > > To clarify that our "risk nodes" are not merely token-level abstractions but rather semantic anchors grounded by the advanced reasoning capabilities of LLMs, we are adding the following paragraph to Section 2.2 (Modeling Vulnerabilities as Divergences at Risk Nodes):
> > >
> > > "It is crucial to distinguish between the optimization level and the representation level of our framework. While TSP applies gradient updates at the token level (optimizing the logits of a specific generation step), the identification and contextualization of a CWE Risk Node are inherently semantic. Real-world vulnerabilities rarely manifest as isolated token errors; they are often the culmination of complex data-flow or control-flow logic. To capture this complexity, TSP leverages the advanced semantic reasoning capabilities of large language models during the automated annotation pipeline (as detailed in Section 3.2). Rather than relying on superficial token matching or rigid heuristics, the annotator LLM analyzes the entire function's context—evaluating control structures, variable scoping, and specific CWE definitions—to isolate the precise root cause within multi-line program logic. Therefore, the target token acts as a 'causal anchor point' for a semantic vulnerability, forcing the model to evaluate the downstream consequences of its localized choices within the broader logic of the program."
> > >
> > > **More thorough and standard limitations analysis(Addressing Q2):**
> > >
> > > We fully agree that a rigorous limitations analysis is crucial for defining our method's operational boundaries. To provide a granular, mechanism-level perspective, we conducted an in-depth attribution analysis based on specific CWE characteristics. By comparing the total vulnerabilities generated by TSP and the Baseline, we explicitly categorized them into regions of superiority and limitation.
> > >
> > > |CWE ID|Vulnerability|TSP|Self-Play|Difference|
> > > |-|-|-|---|-|
> > > |CWE-215|Flask Debug Mode Enabled (Local configuration violation)|4|38|-34|
> > > |CWE-079|Cross-Site Scripting XSS (Local input filtering omission)|16|36|-20|
> > > |CWE-252|Unchecked Return Value (Explicit local check omission)|0|12|-12|
> > > |CWE-119|Buffer Overflow (Local boundary check missing)|20|32|-12|
> > > |CWE-674|Uncontrolled Recursion (Local logic boundary missing)|0|4|-4|
> > >
> > > CWE ID| Vulnerability|TSP|Self-Play| Difference |
> > > |-|-|-|-|-|
> > > |CWE-690|NULL Pointer Dereference (Long-distance state dependency)|14|7|+7|
> > > |CWE-125| Out-of-bounds Read (Cross-procedural/implicit memory operation)|16|11|+5|
> > > |CWE-416|Use After Free UAF (Implicit memory lifecycle)|7|3|+4|
> > > |CWE-073|External Control of File Path (External taint dependency)|4|0|+4|
> > > |CWE-457| Use of Uninitialized Variable (Implicit data flow omission)|15|12|+3|
> > >
> > > TSP excels at handling local features and explicit control flows , significantly reducing or eliminating vulnerabilities like CWE-215, CWE-079, and CWE-252. TSP’s node-level evaluation is highly sensitive to immediate semantic violations (e.g., missing if (!ptr) checks or unsafe configurations). This allows the value network to immediately assign low rewards and prune dangerous branches early in generation.
> > >
> > > Conversely, TSP underperforms on complex memory and implicit data-flow vulnerabilities (e.g., CWE-690, CWE-125, CWE-416). This highlights a core limitation of tree-search reinforcement learning: Delayed State Manifestation. In scenarios like UAF (CWE-416), variable allocation and the final dangerous invocation are separated by long execution distances. Because intermediate steps appear perfectly legitimate, the value estimator suffers from "short-sightedness." Lacking cross-procedural taint-tracking, the model misjudges intermediate safety and fails to foresee the terminal state collapse.
> > >
> > > Your valuable feedback has greatly enhanced our paper's rigor. To thoroughly address this, the reversion will include an independent Limitations section in the main text detailing the CWE analysis. Concurrently, the detailed CWE comparison tables will be added to the Appendix to provide a clear view of our operational boundaries.
> > >
> > > We sincerely appreciate your constructive feedback. Having resolved your concerns and provided the exact revisions to address your remaining points, we hope this fully clears your reservations. In light of these updates, we respectfully ask you to reconsider your overall assessment and champion our work in the final AC discussions.

---

### Decision · Program_Chairs · 2026-04-30

**Decision:**

Accept (regular)

**Comment:**

We thank the authors for their submission, which reviewers found to approach an important problem in a novel and technically reasonable way. The results are also promising, albeit on more modestly-sized LLMs, which is understandable.

The reviewers raised a range of concerns, including missing references, concerns around the task formulation, and reliance on noisy data. The authors responded thoroughly and satisfactorily, making the paper fit for acceptance.